# Effects of individual and organizational factors on job tenure of primary care physicians: A multilevel analysis from Brazil

Ivan Wilson Hossni Dias[1]*, Alicia Matijasevich[1], Giuliano Russo[2], Mário César Scheffer[1,3]

1 Faculdade de Medicine (FMUSP), Departamento de Medicina Preventiva, Universidade de São Paulo, São Paulo, Brasil, 2 Wolfson Institute of Population Health, Queen Mary University of London, London, United Kingdom, 3 Faculty of Medicine of São Paulo University (FMUSP), Brazilian Medical Demography Research Group, São Paulo, SP, Brazil

☉ These authors contributed equally to this work.
* ivanwhd77@gmail.com

## Abstract

### Background

The short tenure of primary care physicians undermines the continuity of care, compromising health outcomes in low-, middle and in high-income countries. The purpose of this study was to investigate the contextual and individual factors associated with the tenure of physician in Primary Health Care (PHC) services. We consider individual-level sociodemographic variables such as education and work-related variables, as well as the characteristics of employers and services.

### Methods

This study is a retrospective cohort study of 2,335 physicians in 284 Primary Health Care Units across the São Paulo, Brazil, public health care system from 2016 to 2020. A multivariate hierarchical model was selected, and an adjusted Cox regression with multilevel analysis was employed. The Strengthening the Reporting of Observational Studies in Epidemiology (STROBE) checklist was used to report the findings from the study.

### Results

The average physician tenure was 14.54 ± 12.89 months, and the median was 10.94 months. Differences between Primary Health Care Units accounted for 10.83% of the variance observed in the outcome, while the employing organizations were responsible for only 2.30%. The physician characteristics associated with higher tenure in PHC were age at hire, i.e., being between 30 and 60 years old, [HR: 0.84, 95% CI: (0.75–0.95)] and professional experience over five years [HR: 0.76, 95% CI: (0.59–0.96)]. Specialties not related to PHC practices were associated with a short tenure [HR: 1.25, 95% CI: (1.02–1.54)].

**Data Availability Statement:** The study includes anonymous information of physicians who worked in primary care services administered by third party organizations. Restriction to sharing de dataset

was imposed by the Research Ethical Committee of the Medical Faculty of São Paulo University. Data from this paper are available upon request to the Ethics Committee of the medical School of the University of São Paulo. Mailing address: 251 Dr. Arnaldo Avenue- Cerqueira César – 01246-000 – São Paulo – SP – Brazil. Phone: + 55 (11) 3893–4401. E-mail:cappesq.adm@hc.fm.usp.br.

**Funding:** This study received support from the Confap-MRC call for Health Systems Research Networks, comprising the following institutions: Newton Fund/ Medical Research Council (UK), Grant Reference MR/R022747/1, Fundação de Amparo à  Pesquisa e ao Desenvolvimento Científico e Tecnológico do Maranhao (FAPEMA-Brazil), COOPI-00709/18 and Fundação de Amparo à Pesquisa do Estado de São Paulo (FAPESP-Brazil), 2017/50356-7. The study also had the contribution of the following research project: ProvMed 2030 – OPAS/MS/FMUSP (Carta acordo n. SCON2020-00001). AM and MCS received support from the National Council for Scientific and Technological Development (CNPq). The funders had no role in study design, data collection and analysis, decision to publish, or preparation of the manuscript.

**Competing interests:** The authors have declared that no competing interests exist.

**Abbreviations:** ASF, Associação Saúde da Família; BMD, Brazilian Medical Demography; CI, Confidence Interval; HRD, Human Resources Database; PHC, Primary Health Care; PHCU, Primary Health Care Units; SPDM, Associação Paulista para o Desenvolvimento da Medicina.

## Conclusion

Differences between Primary Health Care Units and in the individual characteristics, such as specializations and experience, are related to the low tenure of professionals, but such characteristics can be changed through investments in PHC infrastructure and changes in work conditions, policies, training, and human resource policies. Finding a remedy for the short tenure of physicians is essential for guaranteeing a robust PHC system that can contribute to universal, resilient, and proactive health care.

## Introduction

Primary health care (PHC) is expected to save millions of lives and increase the average life expectancy in low- and middle-income countries by 2030 [1]. Current challenges faced by PHC systems globally include responding to the aging of the population, preventing nontransmissible diseases, mitigating risk factors, such as obesity, and preparing for digital transformations and the threat of new pandemics [1,2].

Brazil has stood out as an example of good PHC e.g., by employing family health teams, which contributed to the reduction in the infant mortality rate and hospitalizations for chronic diseases [2]. However, only 65% of the Brazilian population is covered by PHC [2]. The scarcity of physicians and imbalances in their distribution [3], the reduced interest in PHC among recently graduated physicians [4–6], and the short tenure of professionals in these fields [7,8] are all concrete obstacles to the expansion of PHC in Brazil and around the world [9].

The low tenure of physicians in PHC reduces the quality of the assistance provided, breaks the continuity of care given to the population, and consumes the financial resources of the health system [8,10–12]. The percentage of PHC physicians intending to terminate their employment in these fields ranged from 26% in Canada to 70% in China [6]. In San Francisco, 53% of physicians left their work in PHC within two years [8,11], and in São Paulo, 50% of physicians located in the East Zone terminated their employment in the first year after beginning their employment in PHC services [13].

The large number of patients treated [6], burnout [11], poor service infrastructure [14], low salaries [6,14], the need to work multiple jobs, the need to work in both public and private practices [15,16], the privatization of contracts, and the informality of employment relationships [17], in addition to the absence of primary health care content in physician training programs [18], are all factors that have been studied in isolation as possible explanations for the high turnover in PHC.

In the last decade, Brazil has experienced an increase in newly graduate doctors, the majority from private medical schools. Desire for immediate financial gains and the difficulty in accessing specialty training are some aspects that could make these professionals seek PHC services with reduced tenure [19]. Regarding the organizational characteristics, primary care physicians are employed directly by the State or third-party organizations, named Social Organizations [19]. Differences between recruitment politics, selection and incentives applied by these employing organizations are contextual factors that may affect tenure of the physicians in PHC.

Few studies have conducted analysis of the factors that contribute to the tenure of physicians in their service in PHC while considering individual characteristics, training, and work-related variables, including the characteristics of employers and services. *The purpose of this study is to* investigate the contextual and individual factors associated with the tenure of physician in PHC services in the city of São Paulo, the largest urban center in Latin America.

## Material and methods

### Study design, research scenario and inclusion criteria

We conducted a retrospective cohort study of a population of physicians using databases from three private organizations (Social Organizations) that manage the Primary Health Care Services Human Resources Database in the city of São Paulo, in addition to data from the Brazilian Medical Demographics study [20]. This second database includes all physicians registered with the Medical Regional Councils (CRMs), as well as data from the National Commission of Medical Residency (CNRM) and all the medical specialty societies associated with the Brazilian Medical Association (AMB).

The study included 2,335 physicians who worked or were still working in 284 primary health care units in the city of São Paulo from January 1, 2016, to July 17, 2020 (Fig 1). This corresponds to 65% of all physicians who worked in PHC in 2020 in the São Paulo workforce [19]. The Strengthening the Reporting of Observational Studies in Epidemiology (STROBE) checklist was used to report the findings from the study.

The city of São Paulo is a large urban center with approximately 12.2 million inhabitants and an area of $1,171.110$ km$^2$, and 72.7% of its population is covered by 469 Primary Health Care Units [19]. São Paulo accounts for 10.3% of the national gross domestic product [21], has one of the highest concentrations of doctors in the country (4.98/1,000 inhabitants), and is home to 47.3% of all doctors in the state [20]. In 2020, 469 Primary Health Care Units (PHCU) were managed through ten Social Organizations and directly by the State. Three employing organizations concentrated about 65% of the administration of human resources in health. The convenience sampling these three organizations to compose the database for the analysis of this study considered the percentual of the primary care workforce employed and because they allowed access to the anonymous information of physicians to carry out the study.

### Variables, outcome definitions and ethical considerations

To study the tenure of physician employment in PHC services, the following individual variables were chosen: gender, age at hire, city of physician's residence, location of medical school attended, professional experience, medical school type, specialization, weekly workload, and salary. In this study, the variable "workload" refers to the weekly working hours stablished in the contract between the PHC physician and the employer organization. Dichotomizations of the variables 'age at hire', 'professional experience' and 'salary' were applied through the analysis of the distribution of the continuous variables and interquartile intervals.

The PHCUs and the type of employing organization were considered second- and third-level contextual variables, respectively. The definitions and details of the variables are described in detail in Table 1.

The outcome analyzed was *"job tenure"*, *defined as* the difference between the date of initial employment and the date of the physician's termination of contract from PHC up until the date of the closing of the database in July 17st, 2020. The current PHC employment status is the situation of the physician's employment contract within the institution, regardless of whether the termination was the choice of the professional or of the employer. In this study, the termination of contract from PHC is the event of interest of the survival analysis.

A theoretical model that elaborates the hierarchical organization of the individual-level variables was constructed [23]. In this model, the contextual variables related to the services (PHCUs) and to the employing organizations mutually influence the individual-level physician variables. The demographic variables are the most distal determinants of the outcome,

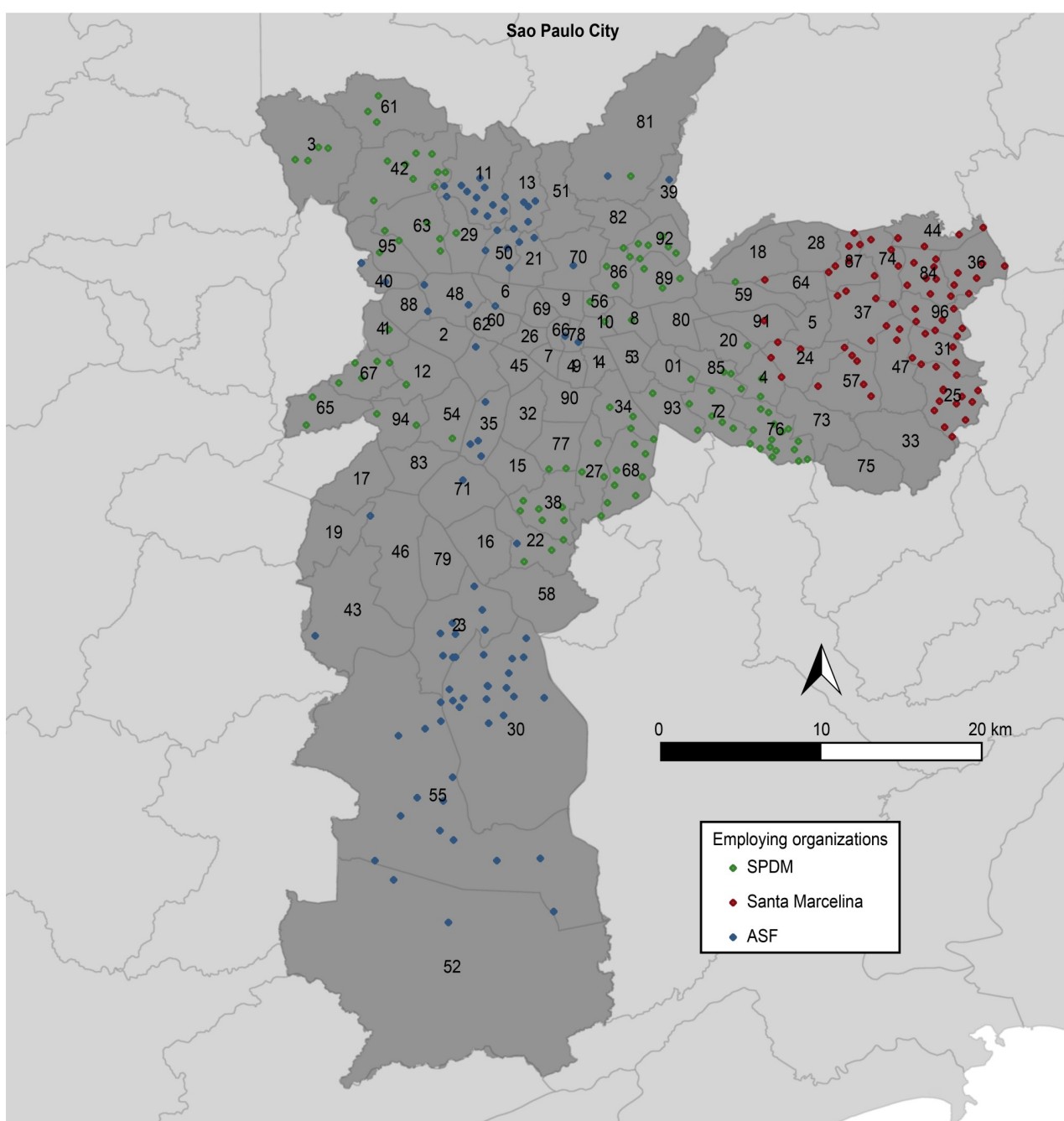

**Fig 1. Distribution of the 284 primary health care units according to the employing organizations.**

having either a direct or indirect influence on the variables associated with the physicians' professional experience and training, which, in turn, can affect the characteristics of their medical employment, such as their workload and salary (Fig 2).

## Data analysis

The descriptive analysis identified the absolute and relative frequencies of the variables studied. Stata v.17 (StataCorp., Texas) and SPSS v.26 software (IBM Corp., Chicago) were used for

**Table 1. Description of the individuals and contextual variables.**

| Level | Variable (source) | Categories | Description |
|---|---|---|---|
| Individual | Gender (HRD) | 0- Female<br>1- Male | Physician gender |
| | Age at hire (HRD) | 1- ≤ 29<br>2- 30– < 60<br>3- ≥ 60 | Physician age at the beginning of the PHC contract |
| | City of physician's residence (BMD) | 1- Sao Paulo<br>2- Other | City associated with the postal code of the physician's home address (City of São Paulo or other municipalities) |
| | Location of medical school attended (BMD) | 1- Sao Paulo<br>2- Other | Municipality where the physician's school/program is located |
| | Professional experience (years) (HRD + BMD) | 1- 1– < 3<br>2- 3– < 5<br>3- 5– < 10<br>4- ≥ 10 | Difference between the date of graduation and the date of initial employment (hiring) in PHC. |
| | Medical school type (BMD) | 1- Public<br>2- Private | Legal nature of the medical program/school from which the physician graduated. Public schools are created or incorporated, maintained and managed by the public authorities and private schools are "maintained and administered by physical persons or through private rights" [22]; |
| | Specialization (BMD) | 1- Family and Community Medicine<br>2- Gynecology and Obstetrics<br>3- Pediatrics<br>4- Nonspecialists<br>5- Other specialty | Registered physician specialty on the initial date of employment in PHC. In Brazil, this requires either completion of a medical residency program or receipt of the title of specialist from a medical society. Nonspecialists are those physicians that have completed medical school but have not received the title of specialist or completed a medical residency. Brazilian legislation allows nonspecialist graduated physicians to work in PHC and other facilities and services. |
| | Current PHC employment status (HRD) | 1- Terminated<br>2- Currently active | Status of the physician's employment contract with the institution. This is the state variable in the survival analysis |
| | Workload (HRD) | 1- 40 hours<br>2. Less than 40 hours | Hours specified in the contract. Variable indicating either a full workload (8 hours a day, 5 days a week) or a partial workload (any workload of fewer than 40 hours a week) |
| | Salary (HRD) | 1- Up to R$6,923.00<br>2- R$6,923–R$10,192<br>3- More than R$10,192 | Initial salary in the local currency at the start of PHC employment divided into three categories |
| Contextual | Employing organization (HRD) | 1- ASF<br>2- SPDM<br>2- Santa Marcelina | The employers of physician's workforce in Sao Paulo. Our study selected the biggest three: ASF: Associação Saúde da Família; SPDM: Associação para o Desenvolvimento Paulista da Medicina e Santa Marcelina |
| | Primary health care unit (PHCU) (HRD) | | The PHCUs in Sao Paulo where the physicians were assigned to work at the time of hire. |

*BMD* Brazilian Medical Demography

*HRD* Human Resources Database

*PHCU* Primary Health Care Unit.

the data analysis. In this study, the data were aggregated to different levels; therefore, we adjusted the Cox multilevel analysis regression by inserting a random effects variable that modifies the hazard rate function at the organization level ($\alpha_j$). This model can be written as:

$$h(t) = h_{0i}(t)\exp(\beta_1 x_{1i} + \beta_2 x_{2i} + \beta_n x_{ni})\exp(\alpha_j)$$

which includes a fixed-effects mixed model ($\beta_1$, $\beta2$, $\beta_n$) and a random effect for group j ($\alpha_j$). The isolated term $\alpha_j$ can be thought of as equivalent to the random intercept coefficient in a linear regression model, while the exponential of $\alpha_j$ affects the frailty term by acting as a multiplier on the baseline risk function [24].

In the analysis of the effect of each variable on the outcome, we applied a significance criterion of 0.20 and used backward elimination [23,25,26]. In this method, a *p value* limit is chosen for the exclusion of explanatory variables (*p-to-remove*), and the analysis is repeated for each

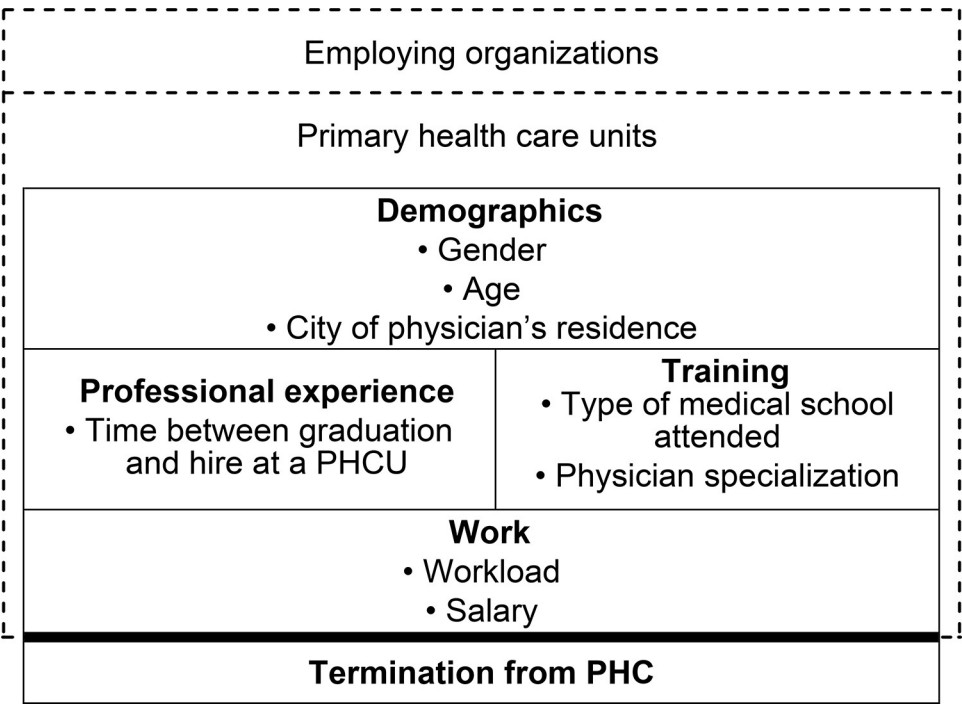

**Fig 2. Conceptual model for the hierarchical analysis of factors influencing physician termination from primary health care.**

hierarchical level. Thus, the model initially includes all variables that are most distal from the outcome and removes any variables with *p value*s that are above the adopted threshold. Once selected, the variables included from each level are kept throughout the regression analysis.

Missing data identified in three variables—location of medical school; professional experience, and type of medical school—were handled by complete case analysis and its occurrence could be explained by the delay to update data of new medical schools on the national databases used in this study.

All methods were performed in accordance with the Declaration of Helsinki and followed all ethical parameters required by the Resolution N° 466/2012 of the National Research Ethics Commission of the National Health Council. The study was approved by the Research Ethics Committees of the Faculty of Medicine of the University of São Paulo (Certificate of Presentation of Ethical Appreciation: 26913419.3.0000.006); from the Municipal Health Department of São Paulo (Certificate of Ethical Appreciation Presentation: 26913419.3.3002.0086) and from Hospital Santa Marcelina under (Ethical Assessment Presentation Certificate: 26913419.3.3001.0066). Prior to the start of the study, written consent and the required administrative permissions were obtained from the Santa Marcelina, SPDM (Associação Paulista para o Desenvolvimento da Medicina) and ASF (Associação Saúde da Família) authorities to access human resource data. Confidentiality and privacy of all physician's data was ensured, used solely and exclusively as a whole for statistical purposes.

## Results

The average tenure of physicians in PHCUs was 14.54±12.89 months, and the median was 10.94 months, with a maximum tenure of 54.41 months (Fig 3). The incidence rate of termination a contract in PHC service was 46.82 [95% CI: 44.33–49.45] per 1.000 persons-month. At

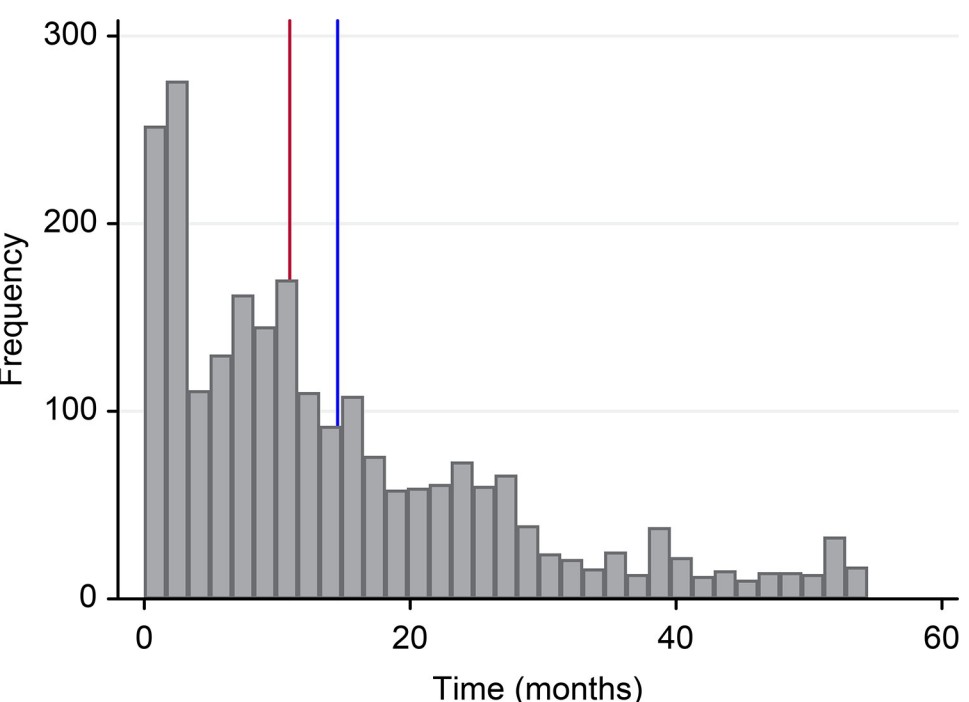

**Fig 3. Length of tenure of physicians in PHC services (Red line = median (10.94), Blue line = mean (14.54); n = 2,335).**

the end of the study, 855 (36.62%) were still actively employed in PHC, while 1,480 (63.38%) had terminated their employment.

Of the physicians in the study, 1,418 (60.73%) were women; 1,172 (50.19%) were between 30 and 60 years old and 1,093 (46.81%) physicians were 29 years old or younger (Table 2). Regarding professional experience, 71.42% of professionals had been hired within the first five years after graduation. Of these, 1,063 (53.77%) physicians had been hired within the first two years after graduation.

Of the total number of physicians, 583 (24.97%) were specialists; the remaining 1,752 physicians (75.03%) had no specialized training. Of the 583 (24.97%) specialist physicians, 522 (89.54%) had one specialization, 53 (9.09%) had two specializations, and 8 (1.27%) had three or more specializations. The most frequent specializations were family and community medicine, with 116 (22.22%) physicians; gynecology and obstetrics, 112 (21.45%); psychiatry, with 92 (17.62%); and pediatrics with 90 (17.24%).

Regarding the type of medical education program, 1,238 (61.53%) of physicians graduated from a private medical school and 774 (38.47%) graduated from a public medical school. A total of 418 (20.77%) physicians graduated from "new" private schools that had been established for less than ten years and the majority of physicians (67.62%) graduated from schools located outside the city of São Paulo.

In our sample of PHC physicians, 1,184 (50.71%) worked full time in PHC (40 hours per week/FTE) and 1,151 (49.29%) worked less than full time (less than 40 hours per week). Among the physicians who worked a 40-hour week, 674 (56.95%) were women, and approximately 67.60% had graduated 1 or 2 years ago. Among the 1,151 physicians working part time, 744 (64.64%) were women, and approximately 59.45% had more than three years of professional experience.

**Table 2. Distribution of physician characteristics in the study (n = 2,335).**

| Variable | Category | N | % |
|---|---|---|---|
| Gender (n = 2,335) | M | 917 | 39.27 |
| | F | 1,418 | 60.73 |
| Age at hire (n = 2,335) | ≤ 29 | 1,093 | 46.81 |
| | 30 –< 60 | 1,172 | 50.19 |
| | ≥ 60 | 70 | 3.00 |
| Location of the medical school attended, city** (n = 1,989) | Other | 1,345 | 67.62 |
| | São Paulo | 644 | 32.38 |
| Location of the medical school attended, state** (n = 2,012) | Other | 987 | 49.06 |
| | São Paulo | 1,025 | 50.94 |
| Professional experience*** (n = 1,977) | 1– < 3 years | 1,063 | 53.77 |
| | 3– < 5 | 349 | 17.65 |
| | 5– <10 | 214 | 10.82 |
| | ≥10 | 351 | 17.75 |
| Type of medical school attended** (n = 2,012) | Private | 1,238 | 61.53 |
| | Public | 774 | 38.47 |
| Specialization (n = 2,335) | Family and Community Medicine | 123 | 5.27 |
| | Gynecology and Obstetrics | 120 | 5.14 |
| | Pediatrics | 102 | 4.02 |
| | Nonspecialists | 1,752 | 75.03 |
| | Other specialty | 238 | 10.19 |
| Employing organization (n = 2,335) | ASF | 845 | 36.19 |
| | SPDM | 978 | 41.88 |
| | Santa Marcelina | 512 | 21.93 |
| Workload (n = 2,335) | 40 hours | 1,184 | 50.71 |
| | < 40 hours | 1,151 | 49.29 |
| Current PHC employment status (n2,335) | Terminated | 1,480 | 63.38 |
| | Currently active | 855 | 36.62 |
| Salary (n = 2,335) | Up to R$6,923.00 | 587 | 25.10 |
| | R$6,923–R$10,192 | 537 | 23.00 |
| | More than R$10,192 | 1,211 | 51.90 |

*ASF* Associação Saúde da Família

*PHC* Primary Health Care

*SPDM* Associação para o Desenvolvimento Paulista da Medicina.

*Missing data: ≤ 10%

**Missing data

Between 10 and 15%

***Missing data: > 15%.

Differences in workload by specialization were also observed. Among the 1,752 (75.03%) physicians without specialist training, 59% worked a full-time schedule; among the specialists, 75% worked a part-time schedule.

Regarding the contextual factors, the unadjusted multilevel analysis was used to measure the presence of random effects in the location and employment relationship. The PHCUs were responsible for approximately 10.83% of the variance in the outcome, while the employing organizations (hirers) were responsible for 2.30%. It is noteworthy that in the adjusted model, there was no reduction in the PHCUs and employing organization random effects in either analysis.

**Table 3. Adjusted analysis for sociodemographic characteristics, qualifications, and professional experience associated with physician termination from PHC (n = 1,977).**

| Variable | Model I | | Model II | |
|---|---|---|---|---|
| | Hazard ratio (95% CI) | p value | Hazard ratio (95% CI) | p value |
| Level 1* | | | | |
| Age at hire | | **0.187** | | **0.187** |
| ≥ 60 | 1.05 (0.76–1.45) | | 1.05 (0.76–1.45) | |
| 30– < 60 | 0.84 (0.75–0.95) | | 0.84 (0.75–0.95) | |
| ≤ 29 | 1 | | 1 | |
| City of physician residence | | **<0.001** | | **<0.001** |
| São Paulo | 0.78 (0.69–0.88) | | 0.78 (0.69–0.88) | |
| Other | 1 | | 1 | |
| Level 2** | | | | |
| Length of time between graduation and hire | | **0.075** | | **0.075** |
| ≥ 10 | 0.85 (0.67–1.07) | | 0.85 (0.67–1.07) | |
| 5– < 10 | 0.76 (0.59–0.96) | | 0.76 (0.59–0.96) | |
| 3– < 5 | 1 | | 1 | |
| 1– < 3 | 1.32(1.10–1.58) | | 1.32(1.10–1.58) | |
| Specialization | | **0.005** | | **0.005** |
| Gynecology/obstetrics | 0.84 (0.62–1.13) | | 0.84 (0.62–1.13) | |
| Pediatrics | 1.22 (0.92–1.63) | | 1.22 (0.92–1.63) | |
| Family/community medicine | 0.79 (0.60–1.05) | | 0.79 (0.60–1.05) | |
| Other specialty | 1.25 (1.02–1.54) | | 1.25 (1.02–1.54) | |
| Generalist | 1 | | 1 | |
| Level 3 | | | | |
| **Workload** | | | | |
| 40 hours | 1.05 (0.91–1.20) | 0.449 | | |
| < 40 hours | 1 | | | |
| Salary (R$) | | | | **0.124** |
| Up to R$6,923.00 | | | 1.12 (0.96–1.31) | |
| R$6,923–R$10,192 | | | 0.93 (0.79–1.11) | |
| More than R$10,192 | | | 1 | |
| Random effects | | | | |
| Employing organization | 0.05 (0.007–0.33) | | 0.05 (0.007–0.33) | |
| Primary health care unit | 0.17 (0.09–0.30) | | 0.17 (0.10–0.31) | |

CI: Confidence Interval.

*Excluding the variable "Gender"

**Excluding the variable "Type of medical school".

In the selection of sociodemographic variables, the age and the current city of residence of those physicians who terminated their employment at a PHCU were associated with job tenure in PHC (Table 3). Being between 30 and 60 years of age at hire [HR: 0.84, 95% CI: (0.75–0.95)] and living in the city of São Paulo [HR: 0.77, 95% CI: (0.68–0.88)] were found to protect against termination of contract from PHC service.

In the analysis of specialized training and professional experience, adjusted for "age at hire" and "city of physician residence", the following variables were associated with job tenure: specialization and the professional experience. Concerning the first, specialties not related to PHC were found to increase the risk of the termination of contracts in PHC [HR: 1.25, 95% CI:

(1.02–1.54)]. Collinear to age at hire, professional experience was significant, and those having more than three years since graduation from medical school [HR: 0.76, 95% CI: (0.59–0.96)] had a lower risk of termination their contracts in PHC.

The contractual variables 'workload' and 'salary' were analyzed separately because they are correlated with each other in the same hierarchical level. The first was not associated with physician tenure and the second was kept in the final model (p = 0.124).

## Discussion

Approximately half of the physicians in São Paulo appeared to have terminated their employment within approximately 11 months of their hire at a PHCU, which is consistent with the findings of other studies showing that the short tenure of physicians is a global phenomenon, which may be detrimental to PHC organizations [4,6,8,27]. The physicians with the highest risk of termination their contract in the PHCUs in the city of São Paulo are young people, under 30 years of age; newly graduate; with less than three years of training in the contract and with specialization not related to PHC.

The present study builds on and develops in multiple ways what is known on job tenure of PHC physicians in Brazil's Southeast [13]. First, it shows that differences between PHCUs affect the tenure of primary care physicians. Our study identifies that type of specialization—Family and Community Medicine and Gynecology or Obstetrics in particular—can be protective factors against high turnover in PHC. In contrast to what found by Bourget et al (2020) [13], our results show that the variable 'workload' was not associated with job tenure, suggesting the existence of a more relevant association with the professional trajectory of PHC physicians who in most cases will actively seek specialization in the first three years of graduation in a hospital setting.

Differently from Bourget et al (2020), our data appear to show that those physicians originally from the district of their current PHC employment, have a lower risk of termination of contract, confirming the importance of physician economic and regional characteristics for tenure in primary care employment [28].

In the city of São Paulo, the PHC medical workforce is mainly composed of young women who have recently graduated and have not studied for or completed a medical specialization.

In the present study, gender was not associated with the risk of termination, although women made up the majority of the study sample. Studies have tried to measure the repercussions of the feminization of medicine for PHC [29–31], considering that women physicians tend to have lighter workloads in PHC but spend more time in appointments with their patients, promoting more robust patient-centered practices compared to those of men [30,32–34].

Approximately 47% of the physicians studied were under 30 years old when hired at a PHCU. In addition to the impact on turnover—our model shows that older physicians are at a lower risk of leaving PHC services—it is possible that young professionals, with their small amount of professional experience and lack of specialization, may partially compromise certain aspects of PHC that are often expected, such as continuity of care and the resolution of a large portion of the health problems faced by the assisted population. The short tenure of young physicians at PHCUs must be considered in assessments of the effectiveness of policies to expand the number of physicians in PHC through the establishment of new medical school programs, as has been implemented in Brazil in recent years.

In Brazil, the number in first-year medical students increased from 20,522 in 2013 to 37,346 in 2019, with 84% due to private schools, alerting to its capacity to train physicians who can work in PHC [20]. Most physicians studied had graduated from a private medical school,

many of which had been established less than 10 years ago and had trained few classes. The physicians graduated from 159 different medical schools, which are heterogeneous in their structure, quality of teaching, and age.

Although graduation from public vs. private medical schools is not associated with physician tenure in PHC services, and previous studies have not found a relation between the type of school and the choice to work in a PHCU [5]. In Brazil and India, graduates of private schools have been shown perform worse than graduates of public schools in national evaluations [35–38]. Likewise, in the countries and regions where a robust PHC system is observed, such as Eastern Europe, Canada, and Australia, most medical schools are public [39].

It is also worth questioning whether the curriculum guidance for medical schools, which very often focuses on surgery and specialized areas, is far removed from PHC and the most urgent health needs of the population [35,37,40,41]. Although curricular content may encourage students to choose to serve in PHC in the future [18,42], PHC is still underrepresented in medical school curricula [43].

Regarding professional experience, approximately 70% of our physicians had graduated less than five years before they were hired at a PHCU. Good medical practices in PHC may be associated with greater physician experience [44]; however, physicians with many years of practice (10–20 years) may be much less inclined to follow care protocols, which are common in PHC, and to update their knowledge [45–47].

Our study shows that specialization is associated with job tenure. This is a cause for concern in Brazil, as there are few family and community medicine physicians in PHC (only 5% of studied physicians), and the PHC workforce is mainly characterized by nonspecialists (75% of studied physicians). What should be considered the best medical qualification for PHC is currently the subject of a worldwide debate [2,9]. In countries such as Canada and Australia, family physicians compose half of the medical workforce currently active in PHC [39].

Although certain specialties, such as family medicine, pediatrics, gynecology and obstetrics, and medical clinic specialties, are related to PHC needs, it is also acknowledged, given the specific competencies and skills that PHC practices require [45,48–50], that continued medical training is important to the provision of high-quality PHC [45–47,49]. Therefore, joint action is necessary to increase the number of family medicine specialists in PHC, as well as to give non-specialist physicians or specialists in areas other than family medicine qualifications to work in PHC.

In the hierarchical model proposed in our study, young, recently graduated, and nonspecialized physicians are those who, when looking for work in PHC, temporarily accept greater workloads and higher salaries only to terminate their employment and move on to become specialized shortly thereafter. This could be a reason why physicians from specialties not related to PHC are at greater risk of terminating their employment and why salary but not workload was associated with physician tenure in PHC, a finding which differs from that of other studies that point to a low level of association [13,14,27].

In the adjusted analysis, living in the capital city of São Paulo proved to protect against termination for physicians in PHC. More opportunities for work, a higher market concentration, and the large number of physician and specialist training centers located in São Paulo may serve as incentives for professionals to continue to work in PHC in the city, as seen in other international contexts [3,5,15].

Physician satisfaction, an aspect closely related to physician tenure and workload in PHC services, is known to vary with the characteristics of the employer and service manager, the location of the PHCU, the level of violence in the community, the demand for care, working conditions, burnout, and the integration of the PHCU with secondary and tertiary reference networks [4,6,7,11,27].

A relevant contribution of this study is that we show that differences between primary health care units accounted for more than 10% of the variance in physician employment termination from PHC, although the employing organizations (hirers) were responsible for only 2.3%. This result suggests similarities in the relationship between contractual conditions and employment among PHC physicians in the city of São Paulo regardless of the hiring organization. This result supports the hypothesis that certain PHC service contexts and characteristics, for instance, their location in neighborhoods or districts closer to the city outskirts and with lower social indicators, are associated with higher physician turnover, which requires further study.

We acknowledge the limitations of our study regarding the use of secondary data from different managing institutions, whose databases can be incomplete and vary in their classification of professionals currently active in PHC. Although the participants in the study represent 65% of the entire PHC workforce of interest, i.e., physicians providing services in all regions of the city, it is worth noting the possible biases caused by the exclusion of one-third of all physicians. For instance, because the excluded physicians were hired by organizations other than the three employers studied, they may be subject to different human resources policies, which could, in theory, have an effect on their tenure in PHC service.

How applicable are our estimates to other PHC settings? This is an important question because different regions and human resources policies can affect the physician turnover. We addressed this issue by inserting contextual factors that generalized best to modelling physician tenure, allowing us to identify the effect of individual characteristics simultaneously with those PHC contextual factors.

Finally, physicians must be evaluated according to factors beyond those considered in the current study, even though those considered here are relevant to the implementation of PHC. Such extended evaluations will require new research, including qualitative research, that considers multiprofessional approaches to PHC and other primary care work processes and determinants.

## Conclusions

The short tenure of physicians employed in PHCUs is a recent phenomenon of great magnitude and complexity and is affected by multiple factors, which can compromise the health care system's ability to meet targets for expansion, population coverage, and improvements in the quality of PHC.

This article contributes to the literature in four ways. First, it offers a description of the primary care physician workforce in São Paulo, the largest urban center in Latin America, adding contextual factors such as employing organizations and the location of the PHCU. Second, it adds variables such as experience and specialization of professionals in studies of medical turnover in PHC, both of which can be modified through medical training policies. Third, its hierarchical model helps to formulate the professional pathway of the physician who works in PHC, a valuable feature for planning the medical workforce in PHC. Finally, it encourages inserting the phenomenon of the high turnover in the policies for providing doctors, considering the characteristics of PHCU, an aspect previously not dimensioned in the scope of PHC.

Our study concluded that the contextual aspects of PHC services and physician characteristics such as age, location of residence, and specialization, are connected to the high turnover of PHC professionals in Sao Paulo, the most populous city in Brazil. These are factors that could be changed through investments in service infrastructure and work conditions and through the implementation of teaching and human resources policies that promote the training, career planning, and financial incentives needed for permanence in PHC.

The solution to the problem of scarcity or short tenure among physicians in primary care is fundamental for enabling PHCs to be integrated into health systems that are more resilient and proactive during health crises and epidemics and that can ensure continuous and universal access to service and better population health outcomes.

## Author Contributions

**Conceptualization:** Ivan Wilson Hossni Dias, Alicia Matijasevich, Giuliano Russo, Mário César Scheffer.

**Data curation:** Ivan Wilson Hossni Dias, Alicia Matijasevich, Giuliano Russo, Mário César Scheffer.

**Formal analysis:** Ivan Wilson Hossni Dias, Alicia Matijasevich, Giuliano Russo, Mário César Scheffer.

**Funding acquisition:** Ivan Wilson Hossni Dias, Alicia Matijasevich, Giuliano Russo, Mário César Scheffer.

**Investigation:** Ivan Wilson Hossni Dias, Alicia Matijasevich, Giuliano Russo, Mário César Scheffer.

**Methodology:** Ivan Wilson Hossni Dias, Alicia Matijasevich, Giuliano Russo, Mário César Scheffer.

**Project administration:** Ivan Wilson Hossni Dias, Alicia Matijasevich, Giuliano Russo, Mário César Scheffer.

**Resources:** Ivan Wilson Hossni Dias, Alicia Matijasevich, Giuliano Russo, Mário César Scheffer.

**Software:** Ivan Wilson Hossni Dias, Alicia Matijasevich, Giuliano Russo, Mário César Scheffer.

**Supervision:** Ivan Wilson Hossni Dias, Alicia Matijasevich, Giuliano Russo, Mário César Scheffer.

**Validation:** Ivan Wilson Hossni Dias, Alicia Matijasevich, Giuliano Russo, Mário César Scheffer.

**Visualization:** Ivan Wilson Hossni Dias, Alicia Matijasevich, Giuliano Russo, Mário César Scheffer.

**Writing – original draft:** Ivan Wilson Hossni Dias, Alicia Matijasevich, Giuliano Russo, Mário César Scheffer.

**Writing – review & editing:** Ivan Wilson Hossni Dias, Alicia Matijasevich, Giuliano Russo, Mário César Scheffer.

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
