## [Decision Letter · Decision Letter 0]

31 Oct 2022

PONE-D-22-18937The effect of individual and contextual factors on the high turnover of primary care physicians: Findings from a multilevel multivariate analysis of a representative cohort in Sao Paulo, BrazilPLOS ONE

Dear Dr. Hossni Dias,

Thank you for submitting your manuscript to PLOS ONE. After careful consideration, we feel that it has merit but does not fully meet PLOS ONE’s publication criteria as it currently stands. Therefore, we invite you to submit a revised version of the manuscript that addresses the points raised during the review process.

We look forward to receiving your revised manuscript.

Kind regards,

Adrian Loerbroks

Academic Editor

PLOS ONE

Journal Requirements:

This study received support from the Confap-MRC call for Health Systems Research Networks, comprising the following institutions: Newton Fund/ Medical Research Council (UK), Grant Reference MR/R022747/1, Fundação de Amparo à Pesquisa e ao Desenvolvimento Científico e Tecnológico do Maranhao (FAPEMA-Brazil), COOPI-00709/18 and Fundação de Amparo à Pesquisa do Estado de São Paulo (FAPESP-Brazil), 2017/50356-7. The study also had the contribution of the following research project: ProvMed 2030 – OPAS/MS/FMUSP (Carta acordo n. SCON2020-00001). AM and MCS received support from the National Council for Scientific and Technological Development (CNPq)

4. Please ensure that you include a title page within your main document. You should list all authors and all affiliations as per our author instructions and clearly indicate the corresponding author.

6. We note that Figure 1 in your submission contain [map/satellite] images which may be copyrighted. All PLOS content is published under the Creative Commons Attribution License (CC BY 4.0), which means that the manuscript, images, and Supporting Information files will be freely available online, and any third party is permitted to access, download, copy, distribute, and use these materials in any way, even commercially, with proper attribution. For these reasons, we cannot publish previously copyrighted maps or satellite images created using proprietary data, such as Google software (Google Maps, Street View, and Earth). For more information, see our copyright guidelines: http://journals.plos.org/plosone/s/licenses-and-copyright.

Reviewers' comments:

Reviewer's Responses to Questions

**Comments to the Author**

1. Is the manuscript technically sound, and do the data support the conclusions?

Reviewer #1: Yes

Reviewer #2: Partly

Reviewer #3: Partly

2. Has the statistical analysis been performed appropriately and rigorously? 

Reviewer #1: Yes

Reviewer #2: Yes

Reviewer #3: Yes

3. Have the authors made all data underlying the findings in their manuscript fully available?

Reviewer #1: Yes

Reviewer #2: Yes

Reviewer #3: No

4. Is the manuscript presented in an intelligible fashion and written in standard English?

Reviewer #1: Yes

Reviewer #2: Yes

Reviewer #3: No

5. Review Comments to the Author

Reviewer #1: Dear authors,

I was a pleasure reading your manuscript.

It is generally well written with a clear structure, well applied research methods and appropriate conclusions. The topic is very relevant. The language is easy to read in proper English. I found no typos.

I only have a few suggestions:

1. Abstract: Please provide no unclear abbreviations here (PHCUs, PHC …), which have not been introduced before

2. Introduction: It would help, if you could describe in a few sentences how primary health care is organized in Brazil. E.g. in Germany < 1% of these physicians are hired by private companies. Furthermore, once in a practice these physicians usually stay there for their whole working life. I guess the GP system in the English NHS is quite different as well with a far lower turnover.

3. What are the physicians doing after quitting their PHC jobs?

4. Page 8, line 174: Do you know, if the ones who worked parttime often had children/ family to take care of?

I will recommend accepting this article after minor revision. All the best!

Reviewer #2: Globally, there is an increasing worry about health workforce shortages in healthcare settings with LMICs experiencing the severest form. The capacity of the health workforce in these regions is insufficient to meet the population’s health objectives; and this disparity is an important limitation in realizing the health-related SDGs. With it potential financial costs and adverse effect on quality patient care, physician turnover is an important subject of interest. The authors are therefore commended for researching into the contextual issues that influence turnover of physicians.

There are however few issues to consider to improve the manuscript

Title

1. The title can be made concise: “Effect of individual and organizational factor on turnover of PC physicians: A multilevel analysis in Brazil”

Abstract

2. What about middle income countries such as Brazil (page 1 line 7); consider including the happening in middle-income countries too

3. The objective of the study was well stated

4. What’s the significance of reporting the median? Any reason? Page 1 line 20. I recommended using the mean if the assumptions have not been breached

Keywords

5. Arrange keywords alphabetically

Introduction and background

6. Reconcile the use of the term medium (Middle)-income countries page 3 line 38

7. Provide reference for page 3 lines 38-41

8. The import should not be hierarchical analysis but factor that contribute to turnover of physicians

9. Contextual factors as used in the title can be explain in details in the background than what has been done.

10. I recommend adding hypothesis as the authors used regression analysis among variables

Method

Study design, research scenario and inclusion criteria

11. Was the study a retrospective study or retrospective cohort study? Page 3 line 70. Reconsider it. What makes it a cohort study?

12. Did the study consider the data of only physicians who have terminated their contract or those in employment were also considered? Page 4 line 77 to 80. If both categories were considered, why retrospective study? Is it not advisable to consider those who have terminated their employment as actual and those in employment as perceived? The write up in page 4 line 96 to 99 do not match the earlier assertion as the analysis concentrated on physicians who terminated their contract in the period under review

Data analysis

13. At what significance was the analysis done?

14. Page 5 line 124; a significance criterion of 0.20 was applied; any justification for the value?

Results

15. I recommend that the authors present only the key findings by summarising the descriptive results from page 6 line 135 to page 8 line 180

Discussion

16. Did the study consider the variables stated in page 12 line 290 to 293? I recommend the authors stick to the objective of the study

17. Any implications to report for healthcare mangers? Also indicate potential areas for future research

References

18. References well written, I will however recommend that the doi is added to the reference list

Reviewer #3: The effect of individual and contextual factors on the high turnover of primary care physicians: Findings from a multilevel multivariate analysis of a representative cohort in Sao Paulo, Brazil

Comments to the Author(s)

Purpose of the study

This study aims to investigate the job tenure among physicians in primary care (n = 2,335). It seems difficult to me to sum up the contents, because many aspects remain unclear to me, above all, because concepts or variables are named differently throughout the manuscript.

Major comments

1. Whole manuscript: According to the abstract, results show that job tenure is associated with age and professional experience as well as being specialised in primary health care. These results do not seem very relevant to me. Is it not logical that people working longer in a certain position tend to be older and more experienced – simply because they spent more time on their job?

However, in the “Results”-section, the wording is different: “age at the time of hire” (lines 189f.) and “time elapsed since the completion of medical school and hire in a PHCU” (lines 196f.) are identified as protective factors against turnover, amongst others.

a) The two factors seem to be intertwined – people who completed school longer ago may tend to be older when beginning a new job. How did you account for the potential interdependence of the predictors in your statistical model?

b) Please use terms consistently throughout your manuscript. Please make sure that the information in your abstract corresponds to the information in your manuscript

2. Whole manuscript: My major concern – beside the fact that the results do not seem very relevant to me – is that several concepts are mixed in this manuscript. In the title, “turnover” in mentioned. The abstract mainly refers to “tenure”. Only in the conclusion, “turnover” is brought up again.

On the one hand, the outcome of this study is defined the “length of time from the date of hire to the termination of the physician’s employment in PHC services” (line 96). This sounds like “job tenure” (and a continuous variable) to me. On the other hand, in the results, protective factors against termination are identified, regardless of the duration, i.e., the job tenure. This sounds like “turnover” (and a dichotomous variable) to me.

Therefore, my suggestion is to adjust title, abstract, and the whole text (maybe even statistical analyses) completely to the concept of “job tenure” OR to the concept of “turnover” including a definition in the introduction and background.

3. Introduction: Please extend the introduction and background. In this section, you might want to present prior evidence on factors that are associated with job tenure/retention/intention to stay OR turnover behaviour among health care professionals. Before this background, you might want to state more explicitly how your research contributes to the field.

4. Please state a concrete study aim in your manuscript.

Minor comments

5. Section “Variables, outcome definitions and ethical considerations”: In the abstract, you explain: “Human resource data from two distinct databases were used.”

a) As I could not find this information in the methods-section, please insert it there.

b) Please describe who collected the data for the two databases, respectively.

c) Please give more specific information on the variables, e.g., gender: Was there a third option?, What kinds of specialisations were given – or was it an open text field? How was professional experience operationalised?, How was workload operationalised?

d) Please specify if you did some recoding of the variables (e.g., dichotomisation) – and if so, how you did that.

6. Page 5, lines 108-110: Please state the number of the Ethics Committee approval.

7. Page 5, line 114: Please give names of enterprises distributing the software you used.

8. Page 6, line 140: Please explain the term “person-time”.

9. Page 6, line 141: Please revise number “26,113.741 months”.

10. Page 6, lines 141f.: If the outcome of this study is the “length of time from the date of hire to the termination of the physician’s employment in PHC services”, how can participants still be “actively employed in PHC”? Please explain.

11. Table 2: Please explain all abbreviations used in the table in a note below.

12. Table 2: As far as I understand it, “workload” and “weekly working hours” are different concepts.

a) Please change the name of your variable or give a rationale why you operationalise “workload” as “weekly working hours”.

b) Please state if the weekly working hours are the hours specified in the contract or the actual working hours including overtime.

13. In the title, you state that the data stem from a “representative cohort” – please explain how you ensured or checked that your sample is a representative one for all physicians in primary care in your study region.

14. Page 7, lines 149f.: Please name and define all your variables in the “Methods”-section.

15. You might want to start the discussion with a short summary of all your results.

16. Page 10, line 243: You may want to revise the beginning of the sentence.

17. Page 13, lines 308f.: “For instance, because the excluded physicians were hired by organizations other than the three employers studied …” I did not quite catch that fact before. Please explain in your methods why you focused on three employers only.

6. PLOS authors have the option to publish the peer review history of their article (what does this mean?). If published, this will include your full peer review and any attached files.

Reviewer #1: No

Reviewer #2: No

Reviewer #3: **Yes: **Katherina Heinrichs

---

## [Author Response · Author response to Decision Letter 0]

10 Dec 2022

November 15, 2022

Revision of manuscript PONE-D-22-18937

The effect of individual and contextual factors on the high turnover of primary care physicians: Findings from a multilevel multivariate analysis of a representative cohort in Sao Paulo, Brazil

Dear Prof. Adrian Loerbroks

Academic Editor, Plos One

Thank you for the opportunity to submit a revision of our manuscript to the Plos One, and for the insightful comments received. 

This document includes all of our responses to reviewers' comments. 

The study includes anonymous information of physicians who worked in primary care services administered by third party organizations. Restriction to sharing de dataset was imposed by the Research Ethical Committee of the Medical Faculty of São Paulo University. Data from this paper are available upon request to the Ethics Committee of the medical School of the University of São Paulo. Mailing address: 251 Dr. Arnaldo Avenue- Cerqueira César – 01246-000 – São Paulo – SP – Brazil. Phone: + 55 (11) 3893–4401 

Regarding the image presented in Figure 1, the map comes from an open-source public institute (Brazilian Institute of Geography and Statistics - IBGE) that specifically allows reproduction to publish these figures specifically under the CC BY 4.0 license. Please note IBGE background maps have bee used by researchers in Brazil for the last 30 years with no property rights restriction.

Please do not hesitate to contact me if you have further comments or questions.

Sincerely,

Ivan Wilson Hossni Dias

Corresponding author

Editors’ comments

Our response: We change file naming, including the separate file labeled 'Revised Manuscript with Track Changes' and the revised paper without tracked changes as a separate file labeled 'Manuscript'. We adjusted the headings with Level 1 18pt font.

Our response: We included in Funding Statement that "The funders had no role in study design, data collection and analysis, decision to publish, or preparation of the manuscript." (Lines 400-402)

3. We note that you have indicated that data from this study are available upon request. PLOS only allows data to be available upon request if there are legal or ethical restrictions on sharing data publicly. For more information on unacceptable data access restrictions, please see http://journals.plos.org/plosone/s/data-availability#loc-unacceptable-data-access-restrictions. In your revised cover letter, please address the following prompts:

Our response: The study includes anonymous information of physicians who worked in primary care services administered by third party organizations. Restriction to sharing the dataset was imposed by the Research Ethical Committee of the Medical Faculty of São Paulo University. Anonymized data from this paper would be available upon request to the Ethics Committee of the medical School of the University of São Paulo. Mailing address: 251 Dr. Arnaldo Avenue- Cerqueira César – 01246- 000 – São Paulo – SP – Brazil. Phone: + 55 (11) 3893–4401

4. Please ensure that you include a title page within your main document. You should list all authors and all affiliations as per our author instructions and clearly indicate the corresponding author.

 Our response: We included a title page with all authors and their affiliations listed, and actualized information of the third author (Giuliano Russo, Wolfson Institute of Population Health, Queen Mary University of London, Email: <g.russo@qmul.ac.uk>, ORCID: 0000-0002-2716-369X) 

Our response: We included the ethics statement in Methods section (page 7, 3rd paragraph, lines170-180) and deleted from the Ethics Section.

6. We note that Figure 1 in your submission contain [map/satellite] images which may be copyrighted. All PLOS content is published under the Creative Commons Attribution License (CC BY 4.0), which means that the manuscript, images, and Supporting Information files will be freely available online, and any third party is permitted to access, download, copy, distribute, and use these materials in any way, even commercially, with proper attribution. For these reasons, we cannot publish previously copyrighted maps or satellite images created using proprietary data, such as Google software (Google Maps, Street View, and Earth). For more information, see our copyright guidelines: http://journals.plos.org/plosone/s/licenses-and-copyright. 

a. You may seek permission from the original copyright holder of Figure 1 to publish the content specifically under the CC BY 4.0 license. We recommend that you contact the original copyright holder with the Content Permission Form (http://journals.plos.org/plosone/s/file?id=7c09/content-permission-form.pdf) and the following text:

 Our response: The map presented in Figure 1 comes from an open-source public institute (Brazilian Institute of Geography and Statistics - IBGE) that specifically allows reproduction to publish these figures specifically under the CC BY 4.0 license. IBEGE maps are not subject to any copyright protection, and have been extensively used to conduct research in Brazil for the last 30 years – see IBEGE policy on maps sharing here < https://www.ibge.gov.br/en/geosciences/maps/state-maps.html>.

 

Reviewer #1

Dear authors,

I was a pleasure reading your manuscript.

It is generally well written with a clear structure, well applied research methods and appropriate conclusions. The topic is very relevant. The language is easy to read in proper English. I found no typos.

Our response: We thank the reviewer’s positive comments and contributions that improve the quality of the manuscript. 

1. Abstract: Please provide no unclear abbreviations here (PHCUs, PHC …), which have not been introduced before.

Our response: The abbreviation of the Primary Health Care Units was excluded from the abstract. Although, we maintain abbreviation of Primary Health care because of its traditional use in family medicine and public health.

2. Introduction: It would help, if you could describe in a few sentences how primary health care is organized in Brazil. E.g. in Germany < 1% of these physicians are hired by private companies. Furthermore, once in a practice these physicians usually stay there for their whole working life. I guess the GP system in the English NHS is quite different as well with a far lower turnover.

Our response: We thank the reviewer for the important comment. We included a new paragraph (page 4, second paragraph, lines 83-90) that briefly describes the primary care organization in the setting of the study. 

3. What are the physicians doing after quitting their PHC jobs?

4. Page 8, line 174: Do you know, if the ones who worked parttime often had children/ family to take care of?

Our response: We thank the reviewer for the important comment. We cannot examine the marital status of the physicians and what are they doing after quitting their PHC jobs due to limitations of the database. 

 

Reviewer #2

Globally, there is an increasing worry about health workforce shortages in healthcare settings with LMICs experiencing the severest form. The capacity of the health workforce in these regions is insufficient to meet the population’s health objectives; and this disparity is an important limitation in realizing the health-related SDGs. With it potential financial costs and adverse effect on quality patient care, physician turnover is an important subject of interest. The authors are therefore commended for researching into the contextual issues that influence turnover of physicians.

Our response: We thank the reviewer’s positive comments and contributions to improve the quality of the manuscript. 

Title

1. The title can be made concise: “Effect of individual and organizational factor on turnover of PC physicians: A multilevel analysis in Brazil”

Our response: We accept the suggestion for the title to “Effects of individual and organizational factors on turnover of primary care physicians: A multilevel analysis from Brazil”

Abstract

2. What about middle income countries such as Brazil (page 1 line 7); consider including the happening in middle-income countries too

Our response: We included the happening in middle-income countries (page 2, line 30). 

3. The objective of the study was well stated

Our response: We thank the reviewer comment.

4. What’s the significance of reporting the median? Any reason? Page 1 line 20. I recommended using the mean if the assumptions have not been breached

Our response: We prefer to use the median due to the asymmetry of the distribution of the time of physician tenure in the Primary Health Care Units which is right skewed.

Keywords

5. Arrange keywords alphabetically

Our response: We rearranged keywords alphabetically (lines 54-55).

Introduction and background

6. Reconcile the use of the term medium (Middle)-income countries page 3 line 38

Our response: We corrected the term medium to middle-income countries (page 3, line 59).

7. Provide reference for page 3 lines 38-41

Our response: We included the reference to the statement in 1st paragraph, lines 59-62.

8. The import should not be hierarchical analysis but factor that contribute to turnover of physicians

9. Contextual factors as used in the title can be explain in detail in the background than what has been done.

10. I recommend adding hypothesis as the authors used regression analysis among variables

Our response (issues 8,9,10): We included a new paragraph to address these suggestions (page 4, 2nd paragraph) including all new description of contextual factors and add hypothesis with respect to the individual and contextual variables considered in the background of the study as follows:

“In the last decade, Brazil has experienced an increase in newly graduate doctors, the majority from private medical schools. Desire for immediate financial gains and the difficulty in accessing specialty training are some aspects that could make these professionals seek PHC services with reduced tenure [19]. Regarding the organizational characteristics, primary care physicians are employed directly by the State or third-party organizations, named Social Organizations [19]. Differences between recruitment politics, selection and incentives applied by these employing organizations are contextual factors that may affect tenure of the physicians in PHC.” 

Method

Study design, research scenario and inclusion criteria

11. Was the study a retrospective study or retrospective cohort study? Page 3 line 70. Reconsider it. What makes it a cohort study?

Our response: This is a retrospective cohort study without control group. Primary care physician employed by the biggest three Social Organizations in Sao Paulo city working in Primary Health Care Units were accompanied from January 1, 2016, to July 17, 2020. According to epidemiological textbook (Rothman KJ, Greenland S, Lash TL. Modern Epidemiology. 3rd ed. Philadelphia: Lippincott Williams & Wilkins; 2008) this can be considered a cohort study because individuals (primary care physicians) with a common exposure (work in Primary Health Care Units) were followed over time, starting form a well-described inception point.

12. Did the study consider the data of only physicians who have terminated their contract or those in employment were also considered? Page 4 line 77 to 80. If both categories were considered, why retrospective study? Is it not advisable to consider those who have terminated their employment as actual and those in employment as perceived? The write up in page 4 line 96 to 99 do not match the earlier assertion as the analysis concentrated on physicians who terminated their contract in the period under review

Our response: The tenure was calculated from the difference between the date of initial employment and the date of the physician’s termination of contract from PHC up until the date of the closing of the database in December 31st, 2016. We have adjusted the 2nd paragraph, page 6 (lines 131-136) to clarify the outcome and the state variable of survival analysis.

Data analysis

13. At what significance was the analysis done?

14. Page 5 line 124; a significance criterion of 0.20 was applied; any justification for the value?

Our response (issues 13,14): We applied a significance criterion of 0.2 as considered in the reference [26] (Chowdhury MZI, Turin TC. Fam Med Com Health 2020;8:e000262. doi:10.1136/fmch-2019-000262). The authors recommend using a p-value range 0.15-0.20 to guarantee that important variables that may have practical reasoning are not missed.

Results

15. I recommend that the authors present only the key findings by summarizing the descriptive results from page 6 line 135 to page 8 line 180

Our response: We remake the Results, excluding first paragraph (page 8), summarized demographic variables and professional experience in 2nd paragraph (lines 187-191) and aspects of medical education program (page 9, 1st paragraph, lines 199-203).

Discussion

16. Did the study consider the variables stated in page 12 line 290 to 293? I recommend the authors stick to the objective of the study

Our response: We adjusted this paragraph with respect to the study objectives, as modified in page 14, 1st paragraph lines 325-326.

17. Any implications to report for healthcare mangers? Also indicate potential areas for future research

Our response: As pointed in 4th paragraph lines 362-371, our study shows that physician turnover in PHC is closely related to the physician’s career, explaining why contract variables such as ‘salary’ and ‘weekly workload’ do not play a significant role in the tenure of physicians in Primary Health Care Units. Organizational factor, including differences between employing organizations and Primary Health Care Units are relevant and reveals the importance of investments in service infrastructure of Primary Health Care Units that could imply in an increase of tenure in PHC.

References

18. References well written, I will however recommend that the doi is added to the reference list

Our response: We included all doi to the reference list. 

Reviewer #3

 The effect of individual and contextual factors on the high turnover of primary care physicians: Findings from a multilevel multivariate analysis of a representative cohort in Sao Paulo, Brazil

Purpose of the study

This study aims to investigate the job tenure among physicians in primary care (n = 2,335). It seems difficult to me to sum up the contents, because many aspects remain unclear to me, above all, because concepts or variables are named differently throughout the manuscript.

Our response: It is sad the reviewer struggled to identify the key contents in our work. We have now tried to streamline the manuscript to make the key messages more prominent and improve the fluidity of the arguments. 

Major comments

1. Whole manuscript: According to the abstract, results show that job tenure is associated with age and professional experience as well as being specialised in primary health care. These results do not seem very relevant to me. Is it not logical that people working longer in a certain position tend to be older and more experienced – simply because they spent more time on their job?

However, in the “Results”-section, the wording is different: “age at the time of hire” (lines 189f.) and “time elapsed since the completion of medical school and hire in a PHCU” (lines 196f.) are identified as protective factors against turnover, amongst others.

a) The two factors seem to be intertwined – people who completed school longer ago may tend to be older when beginning a new job. How did you account for the potential interdependence of the predictors in your statistical model?

b) Please use terms consistently throughout your manuscript. Please make sure that the information in your abstract corresponds to the information in your manuscript

Our response: Our study investigates variables with potential to influence the tenure of those professionals at the time of hiring of the physician in the Primary Health Care Units. Brazil has experienced and unprecedented opening new medical schools, which lead an increase in offer of young physicians into the labour market. Although this gain in workforce, these professionals are initializing its medical career, and the majority will pass through medical residency programs, outside PHC services. Our model helps to understand the termination of the contracts in PHC as a phenomenon intrinsic the professional career of the physician, guiding public policies to study investments in specializing doctors with characteristics that increase its probability to stay on PHC and training young doctors to work in PHC settings.

We agree with reviewer about collinearity between variables ‘age at hire’ and ‘professional experience’. These variables were assessed in two distinct hierarchical levels, and previous study shows (Bourget et al, 2022 [11]) that collinearity did not affect the interpretability of the model. We also maintained these two variables in the same model because of its importance to understanding physician turnover considering the trajectory of the professional through the services.

As pointed by reviewer, we adjusted the terms and language in whole manuscript.

2. Whole manuscript: My major concern – beside the fact that the results do not seem very relevant to me – is that several concepts are mixed in this manuscript. In the title, “turnover” in mentioned. The abstract mainly refers to “tenure”. Only in the conclusion, “turnover” is brought up again.

On the one hand, the outcome of this study is defined the “length of time from the date of hire to the termination of the physician’s employment in PHC services” (line 96). This sounds like “job tenure” (and a continuous variable) to me. On the other hand, in the results, protective factors against termination are identified, regardless of the duration, i.e., the job tenure. This sounds like “turnover” (and a dichotomous variable) to me.

Therefore, my suggestion is to adjust title, abstract, and the whole text (maybe even statistical analyses) completely to the concept of “job tenure” OR to the concept of “turnover” including a definition in the introduction and background.

Our response: We modified 2nd paragraph page 6 to explain the outcome of the study and the state variable of the survival analysis, as we can see: 

“The outcome analyzed was the difference between the date of initial employment and the date of the physician’s termination of contract from PHC up until the date of the closing of the database in July 17st, 2020. The termination of contract is the situation of the physician’s employment contract within the institution. regardless of whether the termination was the choice of the professional or of the employer. This is the state variable of the survival analysis.”

3. Introduction: Please extend the introduction and background. In this section, you might want to present prior evidence on factors that are associated with job tenure/retention/intention to stay OR turnover behaviour among health care professionals. Before this background, you might want to state more explicitly how your research contributes to the field.

Our response: We presented the factors that are related and relevant to the tenure of primary care physicians in page 4, second paragraph (lines 77-82).

4. Please state a concrete study aim in your manuscript.

Our response: We corrected the aim in the manuscript regarding the objective presented in the abstract of the manuscript (page 4, 3rd paragraph).

Minor comments

5. Section “Variables, outcome definitions and ethical considerations”: In the abstract, you explain: “Human resource data from two distinct databases were used.”

a) As I could not find this information in the methods-section, please insert it there.

Our response: This statement was presented in Page 5, 1st paragraph (lines 99-105).

b) Please describe who collected the data for the two databases, respectively.

Our response: The database which came from employing organization is managed by the human resources sector, who was responsible to grant the data to the authors of the study after ethics approval. The second database, which came from Brazil Demography Study 2020 [20] are managed by National Commission of Medical Residency (CNRM), Brazilian Medical Association (AMB) and all the Medical Regional Councils (CRMs) (page 5, 1st paragraph).

c) Please give more specific information on the variables, e.g., gender: Was there a third option? What kinds of specialisations were given – or was it an open text field? How was professional experience operationalised?, How was workload operationalised?

d) Please specify if you did some recoding of the variables (e.g., dichotomisation) – and if so, how you did that.

Our response (issues c) and d)): The variable “gender” was obtained from the regional database of the human resources sector of the employers, without a third option. For the calculation of the variable “professional experience”, we used the dates of graduation in medicine and the date of hiring of the professional in the PHC service, available in the national database of the study of the Medical Demography of Brazil 2020 [20] and the human resources sector of Social Organizations, respectively.

In the categorization of the “specialization” variable, extracted from the national database of the Medical Demography of Brazil 2020 study (15), physicians who obtained their title of specialist granted by specialty societies through the Brazilian Medical Association or by the Brazilian Medical Association were considered specialists. Completion of Medical Residency programs accredited by the National Commission for Medical Residency (CNRM). (19) Specialization in Family and Community Medicine, Gynecology and Obstetrics and Pediatrics were selected as categories apart from other specialists, given that its practice closely to PHC services in Brazil (13,15).

Dichotomizations of the variables ‘age at hire’, ‘professional experience’ and ‘salary’ were applied through the analysis of the distribution of the continuous variables and interquartile intervals. 

6. Page 5, lines 108-110: Please state the number of the Ethics Committee approval.

Our response: We included all the number of the Ethics Committee approvals (Page 7, 3rd paragraph, Lines 168-178)

7. Page 5, line 114: Please give names of enterprises distributing the software you used.

Our response: We included the names of enterprises of the statistical software used (page 6, 4th paragraph, line 147)

8. Page 6, line 140: Please explain the term “person-time”.

9. Page 6, line 141: Please revise number “26,113.741 months”.

Our response (issues 8 and 9): We modify the term “person-time” (measurement that consider the number of people in the study and the amount of time each person spends in the study - time-at-risk) in the paragraph to indicate the incidence rate of termination the contract in Primary Health Care Units (page 8, 1st paragraph, line 182-183).

10. Page 6, lines 141f.: If the outcome of this study is the “length of time from the date of hire to the termination of the physician’s employment in PHC services”, how can participants still be “actively employed in PHC”? Please explain.

Our response: We modify the paragraph indicating the outcome analyzed was the difference between the date of initial employment and the date of the physician’s termination of contract from PHC up until the date of the closing of the database in July 17st, 2020. 

11. Table 2: Please explain all abbreviations used in the table in a note below.

Our response: We included all abbreviations used in the table (Page 27, line 598-00)

12. Table 2: As far as I understand it, “workload” and “weekly working hours” are different concepts.

a) Please change the name of your variable or give a rationale why you operationalise “workload” as “weekly working hours”.

b) Please state if the weekly working hours are the hours specified in the contract or the actual working hours including overtime.

Our response: We standardize the variable to ‘weekly workload’, which correspond the hours specified in the contract of the physician.

13. In the title, you state that the data stem from a “representative cohort” – please explain how you ensured or checked that your sample is a representative one for all physicians in primary care in your study region.

Our response: Using data (probability of observing the outcome in four years = 84,2% and HR adjusted in model VI) from reference [13] we estimate the power cox function in Stata Software (Significance level of 0.05 and power 0.8, two-sided test) which gave us a sample size of 1,387 individuals. Although we have removed this statement form the title, our study includes 65% from the Primary Health Care workforce, working in all different regions in Sao Paulo city, corresponding a 1,977 individual observed. 

14. Page 7, lines 149f.: Please name and define all your variables in the “Methods”-section.

Our response: We corrected and adjusted all the variables naming in Methods section (Page 6, 1st paragraph, line 124-129) and its definitions was presented in Table 1 (lines 128-129).

15. You might want to start the discussion with a short summary of all your results.

Our response: We remake the Results, excluding first paragraph (page 8), summarized demographic variables and professional experience in 2nd paragraph (lines 189-193) and aspects of medical education program (page 9, 1st paragraph, lines 199-203).

16. Page 10, line 243: You may want to revise the beginning of the sentence.

Our response: We corrected the sentence and the beginning the paragraph (page 12, 1st paragraph), as follows:

In Brazil, the number in first-year medical students increased from 20,522 in 2013 to 37,346 in 2019, with 84% due to private schools, alerting to its capacity to train physicians who can work in PHC [19].

17. Page 13, lines 308f.: “For instance, because the excluded physicians were hired by organizations other than the three employers studied …” I did not quite catch that fact before. Please explain in your methods why you focused on three employers only.

Our response: We add this information in a new paragraph (page 5, 3rd paragraph, lines 115-121). We selected theses three employers considering the percentual of the primary care workforce employed and because they allowed access to the anonymous information of physicians to carry out the study.

---

## [Decision Letter · Decision Letter 1]

6 Mar 2023

PONE-D-22-18937R1Effects of individual and organizational factors on turnover of primary care physicians: A multilevel analysis from BrazilPLOS ONE

Dear Dr. Hossni Dias,

Thank you for submitting your manuscript to PLOS ONE. After careful consideration, we feel that it has merit but does not fully meet PLOS ONE’s publication criteria as it currently stands. Therefore, we invite you to submit a revised version of the manuscript that addresses the points raised during the review process.

We look forward to receiving your revised manuscript.

Kind regards,

André Ricardo Ribas Freitas

Academic Editor

PLOS ONE

Reviewers' comments:

Reviewer's Responses to Questions

**Comments to the Author**

1. If the authors have adequately addressed your comments raised in a previous round of review and you feel that this manuscript is now acceptable for publication, you may indicate that here to bypass the “Comments to the Author” section, enter your conflict of interest statement in the “Confidential to Editor” section, and submit your "Accept" recommendation.

Reviewer #2: All comments have been addressed

Reviewer #3: (No Response)

2. Is the manuscript technically sound, and do the data support the conclusions?

Reviewer #2: Yes

Reviewer #3: Partly

3. Has the statistical analysis been performed appropriately and rigorously? 

Reviewer #2: Yes

Reviewer #3: I Don't Know

4. Have the authors made all data underlying the findings in their manuscript fully available?

Reviewer #2: Yes

Reviewer #3: No

5. Is the manuscript presented in an intelligible fashion and written in standard English?

Reviewer #2: Yes

Reviewer #3: No

6. Review Comments to the Author

Reviewer #2: The researchers have addressed all the comments raised during the review. For example, the title fit well the manuscript.

Well done.

Reviewer #3: Effects of individual and organizational factors on turnover of primary care physicians: A multilevel multivariate analysis from Brazil

Comments to the Author(s)

Purpose of the study

This study aims to investigate the job tenure among physicians in primary care (n = 2,335). Organisational as well as individual factors are shown to be associated with job tenure.

The manuscript has improved. However, some of the comments were poorly addressed. Still, there are several issues to be optimised. Line numbers refer to the manuscript without tracked changes.

Mayor comments

1. In lines 131f., you state: “The outcome analyzed was the difference between the date of initial employment and the date of the physician’s termination …”, and in the abstract, you speak of “job tenure”. In the results section, you name your outcome “termination” (e.g., lines 222, 228 and so on). Please – again – use your terms consistently after having defined them. Please consider to add the term “job tenure” in line 131 where you define your outcome (without giving it a proper name), e.g.:

“The outcome analyzed was “job tenure”, defined as the difference between the date of initial employment and the date of the physician’s termination …”

2. Connected to issue #1: If you analyse “job tenure”, you should name it in the title instead of “turnover”. It is a different concept. So, please change the title, e.g.:

Effects of individual and organizational factors on job tenure of primary care physicians: A multilevel multivariate analysis from Brazil

Minor comments

3. Line 39: Please add full stop.

4. Lines 93ff.: The sentence sounds strange. Please consider to rephrase it:

“The purpose of this study is to investigate the contextual and individual factors (…)”

5. Tables 1 and 2 and 3 (!):

a) Please correct category definitions for age; the age of 60 comes up twice:

2- 30–60 -> should be 30– < 60 (?)

3- ≥ 60

b) The same problem occurs with professional experience:

1- 1–2 years -> should be 1– < 3 (?)

What about 2,5 years? Or were only full years stated without decimals? How were the numbers rounded? Please explain in the manuscript.

2- 3–5 -> should be 3– < 5 (?)

3- 5–10 -> should be 5– < 10 (?)

4- More than 10 -> should be ≥ 10 (consistency)

c) The same problem occurs with professional experience. Please correct and use mathematical operators (e.g., instead of “more than”).

6. Table 1 (and WHOLE MANUSCRIPT): Please speak of “weekly working hours” instead of “workload”, because it is a different concept. “Workload” means the amount of work to be done within a particular period of time. But you only refer to the time, not the amount of work (e.g., operationalised as extra hours/overtime). So please change the wording or quote a prior definition that fits to your use of “workload”.

7. You explained in your responses: “Dichotomizations of the variables ‘age at hire’, ‘professional experience’ and ‘salary’ were applied through the analysis of the distribution of the continuous variables and interquartile intervals.” Please give this information in the manuscript too.

8. Line 165: “managed”?

9. Line 177: First mentioning of “SPDM and ASF”? Please explain abbreviations in the text.

10. Table 2 (additional to issues mentioned above):

a) Please use n (lower case) instead of N to describe your sample.

b) Please add commas between definition and next abbreviation, as you did in Table 1.

11. Line 201: Please do not begin a sentence with a number, e.g., “A total of 418 …”

12. Table 3:

a) Please write “analysis” with a lower case a in the Table title.

b) Please use n (lower case) instead of N to describe your sample.

c) Please write “95% CI” instead of “CI 95”.

d) AGAIN, please explain all abbreviations in a footnote (PHC, CI).

13. Line 255: Please delete comma.

14. Conclusions: As far as I know, you do not need to explain your abbreviations again in the conclusions, so please delete the explanations and use the abbreviations.

15. Line 362: Please delete “representative”, as you have not referred to this characteristic of your sample in the manuscript before.

16. Please complete your list of abbreviations.

7. PLOS authors have the option to publish the peer review history of their article (what does this mean?). If published, this will include your full peer review and any attached files.

Reviewer #2: **Yes: **Collins Atta Poku

Reviewer #3: No

---

## [Author Response · Author response to Decision Letter 1]

10 Mar 2023

Reviewer #3

Effects of individual and organizational factors on turnover of primary care physicians: A multilevel multivariate analysis from Brazil

Purpose of the study

This study aims to investigate the job tenure among physicians in primary care (n = 2,335). Organisational as well as individual factors are shown to be associated with job tenure.

The manuscript has improved. However, some of the comments were poorly addressed. Still, there are several issues to be optimised. Line numbers refer to the manuscript without tracked changes.

Our response: We thank the reviewer’s comments and contributions to improve the quality of the manuscript and we hope to address the issues pointed in this review. Line numbers of our responses refer to the manuscript with tracked changes.

Major comments

1. In lines 131f., you state: “The outcome analyzed was the difference between the date of initial employment and the date of the physician’s termination …”, and in the abstract, you speak of “job tenure”. In the results section, you name your outcome “termination” (e.g., lines 222, 228 and so on). Please – again – use your terms consistently after having defined them. Please consider to add the term “job tenure” in line 131 where you define your outcome (without giving it a proper name), e.g.:

“The outcome analyzed was “job tenure”, defined as the difference between the date of initial employment and the date of the physician’s termination …”

2) Connected to issue #1: If you analyze “job tenure”, you should name it in the title instead of “turnover”. It is a different concept. So, please change the title, e.g.:

Effects of individual and organizational factors on job tenure of primary care physicians: A multilevel multivariate analysis from Brazil

Our response: We have now modified the title, changing the term “turnover” to “job tenure” and we adjusted the definition of the outcome, including the term “job tenure” as suggested (page 6, line 136). We also changed lines 140 and 141 to improve the understanding of the event of interest of the survival analysis performed (time-to-event analysis), which is the termination of contract from a PHC service. We changed the term in line 229 and adjusted the first paragraph (lines 229-231), also adjusted to “job tenure” in line 237 (page 10); line 255 (page 11) and line 309 (page 13).

Minor comments

3. Line 39: Please add full stop.

Our response: We modify adding full stop (line 39, page 2). 

4. Lines 93ff.: The sentence sounds strange. Please consider rephrasing it:

“The purpose of this study is to investigate the contextual and individual factors (…)”

Our response: We accept the suggestion and modified the sentence (line 93-94).

5. Tables 1 and 2 and 3 (!):

a) Please correct category definitions for age; the age of 60 comes up twice:

2- 30–60 -> should be 30– < 60 (?)

3- ≥ 60

b) The same problem occurs with professional experience:

1- 1–2 years -> should be 1– < 3 (?)

What about 2,5 years? Or were only full years stated without decimals? How were the numbers

 rounded? Please explain in the manuscript.

2- 3–5 -> should be 3– < 5 (?)

3- 5–10 -> should be 5– < 10 (?)

4- More than 10 -> should be ≥ 10 (consistency)

c) The same problem occurs with professional experience. Please correct and use mathematical operators (e.g., instead of “more than”).

Our response: We changed the mathematical operators of the variables “age” and “professional experience” in all tables. As presented in table 1, the variable “professional experience” was calculated by the difference in years between the date of graduation and the date of initial employment (hiring) in PHC, and in our study it was considered as a continuous variable, allowing decimals in the intervals presented. We also explain the dichotomization process in page 6, lines 129-131.

6. Table 1 (and WHOLE MANUSCRIPT): Please speak of “weekly working hours” instead of “workload”, because it is a different concept. “Workload” means the amount of work to be done within a particular period of time. But you only refer to the time, not the amount of work (e.g., operationalised as extra hours/overtime). So please change the wording or quote a prior definition that fits to your use of “workload”.

Our response: We added a quote a prior adopted in the study to the concept of “workload” (lines 127-129, page 6, first paragraph) as follows:

“In this study, the variable “workload” refers to the weekly working hours stablished in the contract between the PHC physician and the employer organization.”

7. You explained in your responses: “Dichotomizations of the variables ‘age at hire’, ‘professional experience’ and ‘salary’ were applied through the analysis of the distribution of the continuous variables and interquartile intervals.” Please give this information in the manuscript too.

Our response: We have now changed the paragraph that describes the variables of the study and included the information about the dichotomization process of the variables “age at hire”, “professional experience” and “salary” (page 6, lines 129-131). 

8. Line 165: “managed”?

Our response: We changed the term to “handled” (line 171, page 7).

9. Line 177: First mentioning of “SPDM and ASF”? Please explain abbreviations in the text.

Our response: We added an explanation in lines 183-184 (page 8). As presented in table 1, SPDM (Associação Paulista para o Desenvolvimento da Medicina) and ASF (Associação Saúde da Família) are names of the employers organizations of health workforce in Sao Paulo capital. 

10. Table 2 (additional to issues mentioned above):

a) Please use n (lower case) instead of N to describe your sample.

b) Please add commas between definition and next abbreviation, as you did in Table 1.

Our response: We changed the description of the sample to lower case (Table 2) and we added the commas between definition and next abbreviation (line 601 Table 2, page 28)

11. Line 201: Please do not begin a sentence with a number, e.g., “A total of 418 …”

Our response: We modify the sentence as appointed (line 209, page 9)

12. Table 3:

a) Please write “analysis” with a lower-case a in the Table title.

b) Please use n (lower case) instead of N to describe your sample.

c) Please write “95% CI” instead of “CI 95”.

d) AGAIN, please explain all abbreviations in a footnote (PHC, CI).

Our response: We changed Table 3’s title to correct the term “analysis” also presented in line 586 of tables titles descriptions. We changed to lower case de description of the sample (line 603, Table 3). We rewrite the Confidence Interval and add the explanations of the abbreviation in a footnote of the table (line 604).

13. Line 255: Please delete comma.

Our response: We modified the sentence and eliminated the comma (page 11, line 261)

14. Conclusions: As far as I know, you do not need to explain your abbreviations again in the conclusions, so please delete the explanations and use the abbreviations.

Our response: We deleted the explanations (lines 368-371).

15. Line 362: Please delete “representative”, as you have not referred to this characteristic of your sample in the manuscript before.

Our response: We deleted the term from Conclusion (lines 372, page 16).

16. Please complete your list of abbreviations.

Our response: We changed the list of abbreviations adding the employing organization names (ASF Associação Saúda da Família and SPDM: Associação Paulista para o Desenvolvimento da Medicina); CI (Confidence Interval) and the following sources abbreviations (BMD: Brazilian Medical Demography and HRD Human Resources Database)

---

## [Decision Letter · Decision Letter 2]

14 Apr 2023

Effects of individual and organizational factors on job tenure of primary care physicians: A multilevel analysis from Brazil

PONE-D-22-18937R2

Dear Dr. Hossni Dias,

We’re pleased to inform you that your manuscript has been judged scientifically suitable for publication and will be formally accepted for publication once it meets all outstanding technical requirements.

Kind regards,

André Ricardo Ribas Freitas

Academic Editor

PLOS ONE

Additional Editor Comments (optional):

Reviewers' comments:

Reviewer's Responses to Questions

**Comments to the Author**

1. If the authors have adequately addressed your comments raised in a previous round of review and you feel that this manuscript is now acceptable for publication, you may indicate that here to bypass the “Comments to the Author” section, enter your conflict of interest statement in the “Confidential to Editor” section, and submit your "Accept" recommendation.

Reviewer #2: All comments have been addressed

Reviewer #3: All comments have been addressed

2. Is the manuscript technically sound, and do the data support the conclusions?

Reviewer #2: Yes

Reviewer #3: (No Response)

3. Has the statistical analysis been performed appropriately and rigorously? 

Reviewer #2: Yes

Reviewer #3: (No Response)

4. Have the authors made all data underlying the findings in their manuscript fully available?

Reviewer #2: Yes

Reviewer #3: (No Response)

5. Is the manuscript presented in an intelligible fashion and written in standard English?

Reviewer #2: Yes

Reviewer #3: (No Response)

6. Review Comments to the Author

Reviewer #2: (No Response)

Reviewer #3: (No Response)

7. PLOS authors have the option to publish the peer review history of their article (what does this mean?). If published, this will include your full peer review and any attached files.

Reviewer #2: **Yes: **Collins Atta Poku

Reviewer #3: No

---

## [Editor Report · Acceptance letter]

19 Apr 2023

PONE-D-22-18937R2 

Effects of individual and organizational factors on job tenure of primary care physicians: A multilevel analysis from Brazil 

Dear Dr. Hossni Dias:

I'm pleased to inform you that your manuscript has been deemed suitable for publication in PLOS ONE. Congratulations! Your manuscript is now with our production department. 

Kind regards, 

on behalf of

Dr. André Ricardo Ribas Freitas 

Academic Editor

PLOS ONE